# Cryo-EM structure of the polycystic kidney disease-like channel PKD2L1

Qiang Su[1,2,3], Feizhuo Hu[1,3,4], Yuxia Liu[4,5,6,7], Xiaofei Ge[1,2], Changlin Mei[8], Shengqiang Yu[8], Aiwen Shen[8], Qiang Zhou[1,3,4,9], Chuangye Yan[1,2,3,9], Jianlin Lei [1,2,3], Yanqing Zhang[1,2,3,9], Xiaodong Liu[2,4,5,6,7] & Tingliang Wang[1,3,4,9]

PKD2L1, also termed TRPP3 from the TRPP subfamily (polycystic TRP channels), is involved in the sour sensation and other pH-dependent processes. PKD2L1 is believed to be a non-selective cation channel that can be regulated by voltage, protons, and calcium. Despite its considerable importance, the molecular mechanisms underlying PKD2L1 regulations are largely unknown. Here, we determine the PKD2L1 atomic structure at 3.38 Å resolution by cryo-electron microscopy, whereby side chains of nearly all residues are assigned. Unlike its ortholog PKD2, the pore helix (PH) and transmembrane segment 6 (S6) of PKD2L1, which are involved in upper and lower-gate opening, adopt an open conformation. Structural comparisons of PKD2L1 with a PKD2-based homologous model indicate that the pore domain dilation is coupled to conformational changes of voltage-sensing domains (VSDs) via a series of π–π interactions, suggesting a potential PKD2L1 gating mechanism.

[1] Ministry of Education Key Laboratory of Protein Science, Tsinghua University, Beijing 100084, China. [2] School of Life Sciences, Tsinghua University, Beijing 100084, China. [3] Beijing Advanced Innovation Center for Structural Biology, Tsinghua University, Beijing 100084, China. [4] School of Medicine, Tsinghua University, Beijing 100084, China. [5] X-Lab for Transmembrane Signaling Research, Department of Biomedical Engineering and McGovern Institute for Brain Research, Tsinghua University, Beijing 100084, China. [6] School of Biological Science and Medical Engineering, Beihang University, Beijing 100083, China. [7] Beijing Advanced Innovation Center for Biomedical Engineering, Beihang University, Beijing, China 102402. [8] Department of Nephrology, Changzheng Hospital, Second Military Medical University, Shanghai 200433, China. [9] Tsinghua-Peking Center for Life Sciences, Tsinghua University, Beijing 100084, China. These authors contributed equally to this work: Qiang Su, Feizhuo Hu and Yuxia Liu. Correspondence and requests for materials should be addressed to Y.Z. (email: zhangyanqing15@mail.tsinghua.edu.cn) or to X.L. (email: liu-lab@vip.163.com) or to T.W. (email: wangtingliang@mail.tsinghua.edu.cn)

Transient receptor potential (TRP) channels play important roles in a broad spectrum of biological processes, mainly through their extraordinary sensory functions in response to diverse physical, chemical, and biological stimuli[1]. The TRP channel superfamily is classified into seven subfamilies under two groups, namely, group I (exemplified by TRPC, TRPV, TRPM, TRPN, and TRPA) and group II (exemplified by TRPP and TRPML). TRPP subfamily members were identified owing to their sequence similarity to polycystin-1 (PKD1) and polycystin-2 (PKD2). PKD2 together with PKD1 play indispensable roles in autosomal dominant polycystic kidney disease (ADPKD)[2]. Mutations of either PKD1 or PKD2 have been broadly found in patients suffering from the ADPKD. This disease is a potentially lethal monogenetic disorder resulting in major renal manifestations[2].

Polycystin-2 like 1 protein (PKD2L1, also termed TRPP3) and PKD2 (also termed TRPP2) comprise close relatives within the TRPP subfamily (also referred to as the polycystic TRP subfamily)[2], sharing high-sequence similarity (79% homology and 62% identity). They belong to group II TRP channels (TRPP and TRPML), which are characterized by large extracellular domains (exemplified by the polycystic domain and mucolipin domain, respectively) between the first two transmembrane (TM) segment sequences[3–5]. This structural feature is distinct from group I channels such as TRPV or classic voltage-gated ion channels (VGICs)[6]. In addition, PKD2L1 and PKD2 are also members of polycystins/polycystic kidney disease (PKD) proteins. PKD proteins can be classified into two classes: the canonical subtype with six TM domains (6-TM) such as PKD2, PKD2L1, and PKD2L2 and the other subtype with 11 TMs (11-TM) represented by PKD1, PKD1L1, PKD1L2, PKD1L3, and PKD-REJ[7]. The 6-TM subtype is typical for TRPPs and the 11-TM subtype is further characterized by a markedly large extracellular N-terminus, while the final six TM domains of the 11-TM subtype have a structure similar to that of the 6-TM TRPPs[8].

PKD2L1 is encoded by the gene *PKD2L1* (10q25) and was the third member among the PKD proteins to be identified[9]. It is widely expressed in the heart and skeletal muscle, brain, spleen, testis, and retina[10,11]. Even though the deletion of PKD2L1 in murine homologs can induce kidney and retinal defects, a profound understanding of PKD2L1 function remains to be elucidated[10]. Previous studies have confirmed that PKD2L1 is modestly voltage-dependent, regardless of the existence of divalent ions[7,12–14]. Moreover, experiments have shown that PKD2L1 can generate a large tail current in the depolarization–repolarization process, which further indicates the propensity of PKD2L1 to form a nonselective voltage-dependent cation channel[4,7,15]. PKD2L1 is thought to be involved in the formation of functional channel complexes with PKD1L3 or PKD1L1, two homologs of PKD1[16,17]. This PKD1L3/PKD2L1 complex is sensitive to $Ca^{2+}$ impetuses, as well as mechanical stress and acid stimuli through off-responses, among which the acid sensing may be responsible for the function of sour taste and pH-dependent regulations[14,16,18–20]. Additionally, the PKD1L1/PKD2L1 complex can work as a ciliary calcium channel controlling ciliary calcium concentration and thereby modulating hedgehog pathways[17,21].

TRP channels usually contain dual gates (upper and lower) that regulate ion permeation[22]. A few closed-state cryo-electron microscopy (cryo-EM) structures of PKD2 (PDB code: 5T4D, 5K47, 5MKE, and 5MKF at resolutions of 3.0, 4.22, 4.3, and 4.2 Å, respectively) have been published and two hypothetical models have been provided with respect to gating mechanisms[4,23,24]. One model suggests that PKD2 is voltage-gated and the conformational changes within the voltage-sensing domain (VSD) accommodate the opening of the lower gate[4]. The other model

suggests that the conformational changes of polycystin domains are the driving force of the gate opening[23]. However, drawing conclusions from the second model is challenged by its low resolution and the slight dominant orientation of its EM map, which may limit the accuracy for specific side-chains and ions. Of all the members of the PKD family, we found that PKD2L1 is one of mostly widely studied channels that can form a functional homotetrameric channel when heterologously expressed on the plasma membrane[7,17,21]. Because of its considerable importance to PKD protein-related diseases, numerous in vitro and in vivo functional experiments have been performed to discover its mechanisms; however, the intricate regulatory mechanisms of PKD proteins remain to be uncovered.

To understand the structural basis underlying the gating mechanism of PKD2L1, we determine the cryo-EM structure of mouse PKD2L1 with C4 symmetry at an overall resolution of 3.38 Å. Notably, the whole structure of PKD2L1 differs from that of PKD2 (PDB:5T4D)[4]. The selectivity filter/upper gate of PKD2L1 adopts an open conformation compared to that of PKD2. On the basis of our comparison with the PKD2-based homologous model of PKD2L1 structure, we speculated that in PKD2L1, the concurrence of gate opening and VSD conformational changes was caused by coupling between the VSD and the pore domain. Our structural study presented the insight into the structure of PKD2L1 at atomic resolution, which provided a molecular basis for understanding the function of PKD2L1 and expanded the observed conformational landscape of TRPP channels.

## Results

**Cryo-EM structure of PKD2L1.** In order to enhance biochemical stability, we truncated large portions of the N- and C-termini of PKD2L1 to form the PKD2L1 (residues 64–629) construct. This construct also lacks segments of the EF hand[25] and a coiled-coil domain[26], both of which are functionally dispensable according to previous studies[7,26] (Fig. 1a). The PKD2L1 (residues 64–629) construct contains the major functional domains of PKD2L1: (a) the VSD, constituting helices S1–S4, (b) the pore domain constituting helices S5, S6, and the intervening sequence forming the PH1 and PH2 pore helices (PHs), (c) the entire extracellular polycystin domain, which separates the first two TM helices[4], and (d) the oligomerization domain (OD)[7]. To help understand the structures available thus far, we functionally validated our truncated channel (PKD2L1_64–629) with both $Ca^{2+}$-induced response (ICE or influx-operated $Ca^{2+}$ entry, $I_{Ca}$)[26] and acid-evoked current response (off-response, $I_{pH}$)[16], resembling all the major characteristics established from wild-type (WT) PKD2L1. Moreover, the responses of PKD2L1_64–629 indeed exhibited significant increases in amplitudes and time constants of the decay phase (Fig. 1e–h)[26], in agreement with our expectation that PKD2L1_64–629 should be even more active than WT PKD2L1. Therefore, the PKD2L1 structure we achieved is based on PKD2L1_64–629 with confirmed and enhanced functionalities when expressed in mammalian cells[26]. By using FEI Titan Krios transmission electron microscope equipped with a Falcon II detector and a Cs corrector, we resolved the cryo-EM structure at a resolution of 3.38 Å, according to the gold-standard Fourier shell correlation (FSC) 0.143 criterion[27] (Supplementary Figs. 1a–1f and 3, Supplementary Table 1).

The PKD2L1 (residues 64–629) structure is composed of four protomers with unrecognizable EM densities around cytoplasmic domains and allows the visualization of residues 101–563. It is approximately 90 Å in diameter and 85 Å in height (Fig. 1c). The VSDs are formed by the first four TM helices (S1–S4) and the pore domain is composed of four S5-PH-S6 fragments. Due to the

analogous domain organization and high-sequence similarity of PKD2L1 and PKD2, PKD2L1 potentially adopts a swapped feature as well. The polycystin domains lock onto each other on the extracellular side, stabilizing the homotetrameric ion channels (Fig. 1d and Supplementary Fig. 6a). This overall architecture of PKD2L1 is identical to that of PKD2, as anticipated. Nevertheless, the PKD2L1 structure demonstrates remarkably distinct details in comparison with the closed PKD2 structures, with several notable differences between the homologs (PDB code: 5T4D, 5K47, 5MKE, and 5MKF, Supplementary Fig. 5a, c)[4]. One of the key

differences between the structures of PKD2L1 and PKD2 is that the S4–S5 linker in PKD2L1 is invisible at near-atomic resolutions, which reflects the flexibility of the linker (Fig. 1b, c and Supplementary Fig. 5a, c). The PDK2L1 pore domain displays an obvious dilation when viewed from the side (Fig. 2a).

**An open-pore structure.** The pore radii along the permeation pathway of PKD2L1 and PKD2 were calculated by HOLE[28] (Fig. 2a, b). For both channels, cations need to pass through the

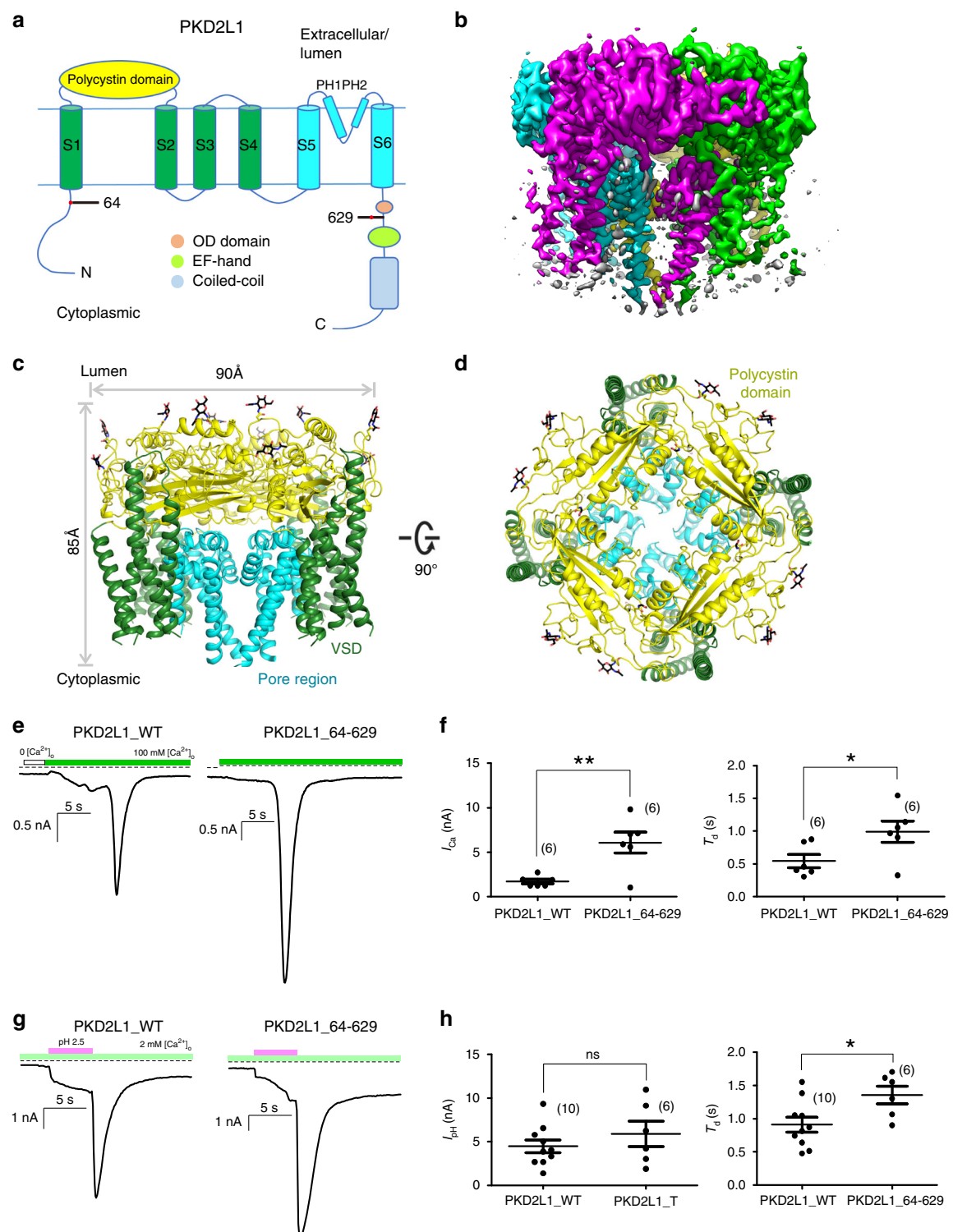

narrow neck, the upper gate, and the lower gate corresponding to the polycystin domains (pre-pore valley), the selectivity filter, and the constriction residues of S6, respectively. These three functional domains sequentially separate the pore space/pathway into extracellular vestibule, internal cavity, central cavity, and intracellular vestibule (Fig. 2a). PKD2L1 channels contain two distinct pore constrictions along the ion permeation pathway (Fig. 2b), similar to TRPV1[29] and PKD2[4].

Structural comparison of the pore domains of PKD2L1 and PKD2 reveals that PKD2L1 has an open-pore structure (Fig. 2b). Two short PHs shape the upper gate featuring the selectivity filter at the outer pore, whereas an S6 helical bundle seals the bottom as the lower gate of the inner pore (Fig. 2c). Located in the outer pore/selectivity filter region, three residues encircle the narrowest constrictions of both PKD2L1 and PKD2: $L^{521}GD^{523}$ (PKD2L1) and $L^{641}GD^{643}$ (PKD2), respectively (Fig. 2d). The diagonally positioned carboxyl groups of D523 (resp. D643 for PKD2) are 10.9 Å (resp. 10.6 Å) apart, which does not show any obvious difference in comparison between proteins. Conversely, diagonal distances between carbonyl group atoms of L521 and G522 for PKD2L1 are 10.4 and 7.3 Å, respectively, whereas the analogous distances for PKD2 (between corresponding residues L641 and G642) are 5.0 and 6.9 Å, respectively. L641 serves as the constriction site (5.0 Å) that prevents $Ca^{2+}$ and $Na^+$ passing through the upper gate at the closed state of PKD2; whereas G522 is the narrowest site (7.3 Å) of the PKD2L1 structure, representing the more permissive state allowing ions to conduct. One of the intensively studied TRP channels TRPV1, having been resolved in different conformations, increases its upper gate from 4.6 to 7.6 Å when changing from the closed state to the fully open state (PDB code: 5IRZ and 5IRX; Supplementary Fig. 5b, Supplementary Table 2)[22]. The narrowest diameter is about 7.3 Å in our PKD2L1 structure, comparable to TRPV1's open-state structure (PDB code: 5IRX), providing further evidence that its upper gate is likely in an ion-conducting state.

Another line of evidence for the open state comes from the lower gate in the S6 helical bundle-crossing region in both PKD2L1 and PKD2 (Fig. 2c, f). In contrast to the major constriction L677 in PKD2 (L557 in PKD2L1), the lower gate of PKD2L1 is presumably enclosed by the constriction I560 (I680 in PKD2, Fig. 2a, b). The diagonal distance of the PKD2L1 I560 site is 8.3 Å (Fig. 2f). L677 serves as the constriction site (4.9 Å) at the closed state of PKD2; whereas I560 is the narrowest site (8.3 Å) of the open state, still allowing hydrated ions to pass through the gate. The comparison of the lower gate between the ion-conducting state (PKD2L1) and the non-conducting state (PKD2) suggests the divergence in S6 helices (Fig. 2c, e). By comparing PKD2L1 with other known structures of open conformation, e.g., Na$_v$Ms, RyR2, and TRPV1 with lower-gate constrictions of 7.8, 8.4, and 9.3 Å, respectively (Supplementary

Table 2)[30–32], we suggest that the lower gate (8.3 Å) also represents an open state in the PKD2L1 structure.

In all, PKD2L1 pore is substantially more dilated, as compared to PKD2 structure at the closed state (Fig. 2). G522 and I560 constitute the constriction of the upper gate and the lower gate of PKDL1, respectively. For PKD2, the residues represented by L677 form the narrowest constriction at the lower gate and L641 is the constriction site of the outer pore.

To gain functional support for the above structural insights into gating, we functionally tested the two constriction sites corresponding to the outer (G522) and lower (I560) gates of PKD2L1, respectively. Mutations G522L or I560F severely impaired PKD2L1 gating, evidenced by loss of function in response to both stimuli ($I_{Ca}$ and $I_{pH}$) (Supplementary Fig. 4), potentially by side-chain extensions of Leu or Phe to occlude $Ca^{2+}$ and $Na^+$.

Interestingly, with respect to the constriction L677 in PKD2, L557 in the PKD2L1 structure does not face the central permeation pathway and thus cannot form the lower gate. For a better interpretation of such difference, we overlaid S6 of PKD2L1 and PKD2. Structural alignment of S6 reveals a π-helix in the middle of S6 in PKD2, whereas S6 adopts a canonical α-helix in PKD2L1 (Fig. 2e). The π-helix is a rare secondary structure derived by the insertion of a single amino-acid into a common α-helix[33]. The presence of a π-helix in PKD2 accompanies the C-terminal part of S6 to bend toward the permeation pathway and simultaneously, introduces one residue shift along the helical axis. This discrepancy in the lower gates of PKD2 and PKD2L1 is observed owing to their differences in secondary structure (Fig. 2e). Altogether, these data suggest that the high-energy π-helical S6 and the low-energy α-helical S6 may represent different functional states of PKD2 and PKD2L1.

Based on the different conformations of the gates in PKD2L1 and PKD2, the structure of PKD2L1 may represent the open conformation conducting cations through the two gates.

**The polycystin domains.** Polycystin domains of both PKD2L1 and PKD2 display a sandwich-like shape with five β-sheets in the tilted middle layer, three α-helices on one side, and a large loop containing two short antiparallel β-sheets on the other side (Supplementary Fig. 6a, b). Each protomer in PKD2 has three glycosylation sites: N375, N362, and N328, corresponding to residues G254, N241, and N207 in PKD2L1 respectively. In the structure of PDK2L1, oligosaccharides are identified on N241 and N207 (Supplementary Fig. 6b). Considering the role of N375 in stabilizing the inter-domain interactions with respect to the PKD2 S3–S4 linker[24], the absence of this specific asparagine and the corresponding oligosaccharide chain in PKD2L1 may impact the interactions between PKD2L1 S3–S4 linkers and its polycystin domains (Supplementary Fig. 6c). Moreover, the density of the

**Fig. 1** Structural characterizations of PKD2L1. **a** General topology of PKD2L1. PKD2 has a similar topology with a longer N-terminus and no oligomerization domain (OD). **b** Overall EM map of PKD2L1 (residues 64–629) colored according to different protomers. A swap structure feature can be visualized. Shown from the side view. **c**, **d** Overall structure of PKD2L1 (residues 64–629), view from the side and top, respectively. The structure is domain-colored as in panel **a**. The same color scheme was applied throughout the structural analysis in Figs. 1–6, Supplementary Figs. 5 and 6, and Supplementary Movies 1–3. Glycosyl moieties are shown as sticks. All the above structure figures were prepared using PyMol (The PyMOL Molecular Graphics System, Version 1.8 Schrödinger, LLC.) and the EM map was generated using UCSF Chimera. **e**, **f** $Ca^{2+}$ response $I_{Ca}$ was elicited when $[Ca^{2+}]_o$ rapidly switched from 0 to 100 mM (green bar). Exemplar traces (**e**) for PKD2L1_WT (left) or PKD2L1_64–629 (right), both in complex with PKD1L3. Statistical summaries of $I_{Ca}$ (**f**) for the peak amplitude, and the time constant of the decay phase ($T_d$). Number of cells for each group are indicated in the parentheses. PKD2L1_64–629 represents the truncated version of PKD2L1: 64–629, from which the Cryo-EM structure was resolved. **g**, **h** Acid response ($I_{pH}$) was induced when the stimulus of pH 2.5 (red bar) was quickly withdrawn. In all, 2 mM $[Ca^{2+}]_o$ was included in the bath solution for $I_{pH}$ throughout this study. Exemplar traces (**g**) for PKD2L1_WT (left) or PKD2L1_64–629 (right), both in complex with PKD1L3. Statistical summaries of $I_{pH}$ (**h**) for the peak amplitude, and the time constant of the decay phase ($T_d$). Intracellular buffer of 0.5 mM EGTA was used throughout. All above values are in mean ± SEM, indicated with significance (*$p < 0.05$; **$p < 0.01$; and ***$p < 0.001$). More detailed studies including the negative control can be found in our previous paper[26]

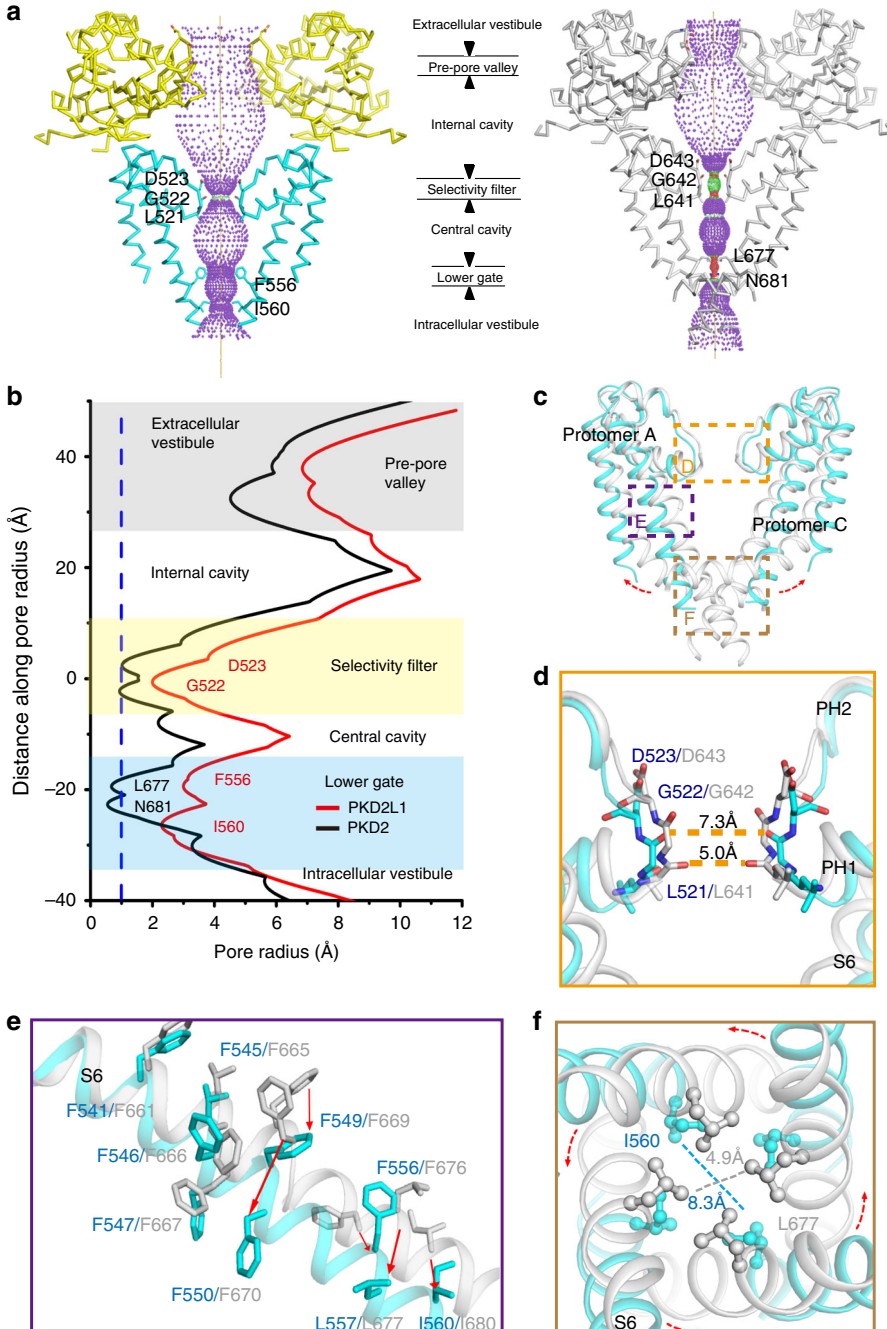

**Fig. 2** PKD2L1 pore suggests its open conformation in comparison with PKD2 structure. Throughout the analysis, PKD2L1 structures are colored in cyan, green, or yellow according to different domains, and PKD2 or closed-state PKD2L1 model are colored in gray. **a** The channel passage of PKD2L1 (left) or PKD2 (right, 5T4D) is indicated by purple dots in the center of two diagonal protomers (pore region from chain A and C, polycystin domain from chain B and D) in each channel (with radius smaller than 1.15 Å colored red, between 1.15 and 2.3 Å colored green). The position of selectivity filter is set as the origin of the y-axis. **b** Pore radii of PKD2L1 (red) and PKD2 (black; 5T4D) along the ion-conducting pathway calculated by HOLE program. **c** Superimposed examinations between the pore domains of PKD2L1 and PKD2 (only two S5-PH-S6 of diagonal protomers are shown). **d** Comparison of the selective filter/ upper gate between PKD2L1 and PKD2. To emphasize the difference, the two structures of PKD2L1 (cyan) and PKD2 (gray) were superimposed relative to pore helices PH1 and PH2. The constriction site at the upper gate is G522 (PKD2L1) or L641 (PKD2), respectively. Corresponding to PKD2L1 residues, the residue names and numbers in PKD2 are shown in gray immediately after the slash, throughout this study when applicable (In the case of closed-state PKD2L1 model, the respective residue names and numbers in PKD2 are shown in gray within parentheses). **e** S6 segments of PKD2L1 and PKD2 adopt different conformations. A high-energy π-helix can be witnessed in PKD2 while PKD2L1 only gets a normal α-helix. The absence of this π-helix in PKD2L1 accompanies the C-terminal part of S6 to bend away from the permeation pathway and to shift one residue towards the cytosol. Red arrows emphasize the discrepancies with respect to side chains of marked residues. **f** Comparison of the lower gate between PKD2L1 and PKD2. Two structures were superimposed similarly as in **c**. The constriction site at the lower gate is I560 (PKD2L1) or L677 (PKD2), respectively. The counterclockwise red arrows emphasize the comparative differences from PKD2 to PKD2L1

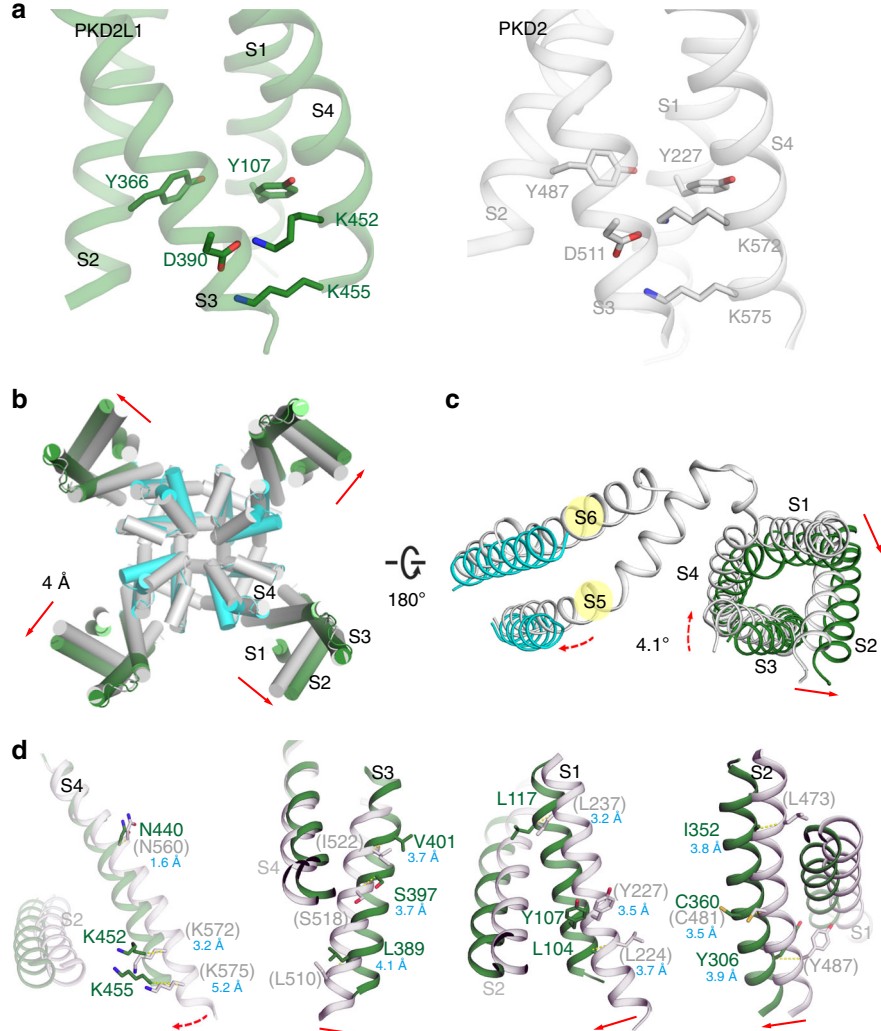

**Fig. 3** Conformational changes of the VSDs modeled in PKD2L1. **a** PKD2L1 (green) and PKD2 (gray) both harbor two lysines, K452 and K455 (resp. K572 and K575 in PKD2) in S4, which increases their voltage sensitivity. The two lysines are potentially stabilized by D390 from S3, Y107 from S1, and Y366 from S2 (resp. D511, Y227, and Y487 in PKD2). **b** Cytosolic view of structural comparison of PKD2L1 (colored as in Fig. 1) and its homologous closed-state model, excluding polycystin domains for the ease of visualization. The whole structure of PKD2L1 displays a clockwise rotation relative to its closed state as viewed from the intracellular side. The red full arrows indicate overall shifts of each protomer from the closed to open state. **c** One protomer from panel **b** viewed from the extracellular side; only six transmembrane helices of this protomer are shown. The red dotted arrows indicate the rotation of specific regions. **d** Dissection of the conformational shifts of the VSD segments relative to the pore domain. The open-state cryo-EM PKD2L1 structure and the closed-state PKD2L1 model are superimposed overall. Adjacent helices are shown in each panel as the reference to indicate the orientation of the structures. The respective residue names and numbers in PKD2 are shown in gray within parentheses throughout this study when applicable (esp. Figs. 3–5)

seven residues between Q296 and E302 in PKD2 is invisible, while it is clearly resolved in PKD2L1 (corresponding residues from W174 to L180). These seven residues form a loop region containing the glycosylation site N177. Hydrogen bonds formed by N176/Q289 and Y175/L294 enhance the interaction of the loop region and the adjacent polycystin domain. Such an organization facilitates the assembly of the polycystin domains in PKD2L1 and maintains their steadiness (Supplementary Fig. 6d).

Meanwhile, the polycystin domains of PKD2L1 share similar structural features to PKD2. Two prominent grooves are defined for intra- and inter-subunit interactions (Supplementary Fig. 6e)[4]. Interestingly, the static nature of such TRPP-type polycystin domains is similar to that of the mucolipin domain of TRPML1, another family member of group II TRP channels. There is no obvious conformational change of the mucolipin domain in response to stimuli of different pH (PDB code: 5TJA, 5TJB,

5TJC)[3]. And this phenomenon can also be observed in ligand-gated TRPML1 channels (PDB code: 5WJ5, 5WJ9)[3,34–37]. This static characteristic of polycystin domains is of considerable significance in understanding the overall synergic and working mechanism of channel assembly.

**The voltage-sensing domains.** PKD2L1 is supposed to be a voltage-dependent channel and possesses a representative VGIC fold, with its first four TM helices forming the VSD[7]. Similar to PKD2 but unlike TRPV1 and TRPA1, PKD2L1 harbors two lysines in S4 (K452 and K455), which could partially explain the voltage-dependent properties. The two lysines are potentially stabilized by residues from S1, S2, and S3 (exemplified by Y107, Y366, and D390) (Fig. 3a). Due to the limited resolution of the EM map near the cytosolic side, we refrain to talk about the detail interactions of this region.

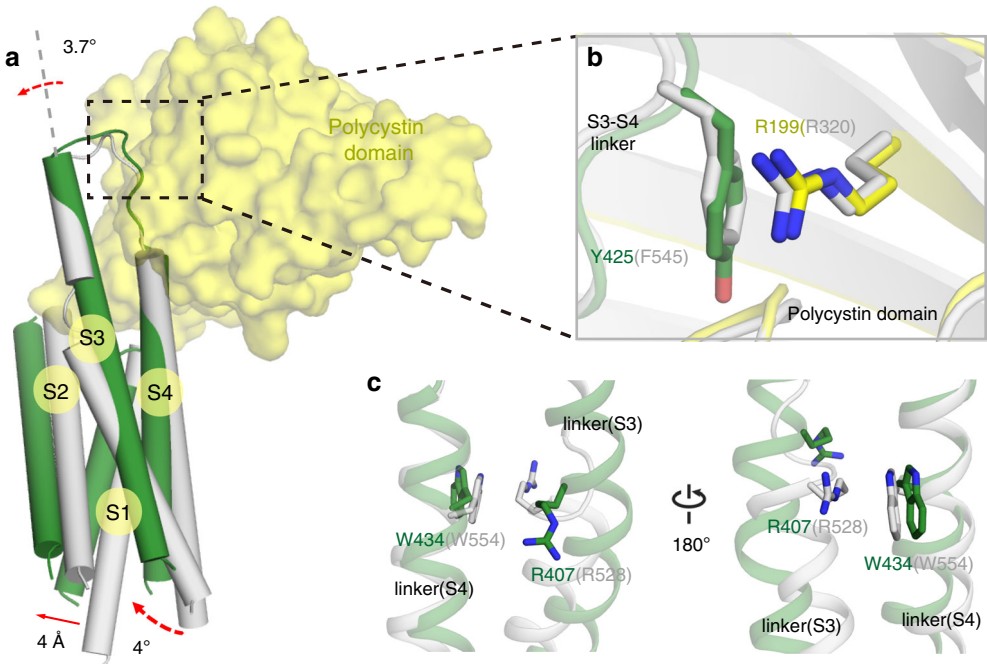

**Fig. 4** Outward movements of the overall VSD modeled in PKD2L1. **a** The VSD domain of PKD2L1 (green) has an outward movement compared to that of the homologous PKD2L1 closed-state model (gray; the modeled structure from PKD2 structure 5T4D), as indicated by red full arrows or dotted arrows. **b**, **c** The helix-turn formed by the PKD2L1 S3–S4 linker appears to exhibit a small-angle rotation (**c**) around a cation-π axis (**b**) by comparing the two PKD2L1 states. Cation–π interactions are formed between R199/Y425 in PKD2L1 and remain stable in both the open-state cryo-EM structure and the closed-state PKD2L1 model (resp. R320/F545 in PKD2). **c** A cation–π interaction within the S3–S4 linker (R407/W434 in PKD2L1; R528/W554 in PKD2) is disrupted during the small-angle rotation from closed to open conformation. This process may change the orientation of S3 and S4 helices and may further effect the overall arrangement of the VSD and pore domains

Although PKD2L1 highly resembles PKD2 both sequentially and topologically, the structure of PKD2L1 TM domains still differs from PKD2 in terms of conformational details. To visualize the conformational changes of each segment between PKD2L1 and PKD2, a homology-based model of closed-state PKD2L1 was generated using the online SWISS-MODEL workspace (https://www.swissmodel.expasy.org)[38].

During the inspection of PKD2L1 conformational changes, we observed that the polycystin domains exhibit no obvious variance, while all VSD helices present a clockwise twist approximately 4° from an intracellular view and show an outward translation of about 4 Å (Fig. 3b and 4a, Supplementary Fig. 1, and Supplementary Movie 1). Specifically, S4 undergoes a 4.1° self-rotation and shifts toward the extracellular side of about half of a helical turn, while the S1, S2, and S3 segments appear to stand approximately parallel to each other in both structures and move like a single module of approximately 4 Å (Fig. 3b, c, d). Among these VSD helices, S3 in the open-state cryo-EM PKD2L1 structure exists as an entire helix connecting to the S3–S4 linker without kinks, whereas in the closed-state PKD2L1 model (as well as the PKD2 structure) it is separated by a piece of loop fragment (S3-kink). Additionally, the S3–S4 linker, in the form of a helix-turn, appears to undergo a notable small-angle rotation (roughly 3.7°) around the R199/Y425 cation-π axis (Fig. 4a, b). Transition from the closed state to the open state disrupts the cation-π interaction between R407/W434 (resp. R528/W554 in PKD2) within the S3–S4 linker (Fig. 4c). This phenomenon together with the S3-kink disappearance implies different functional states of PKD2L1. These details of conformational changes in the VSD are critical in depicting the molecular mechanism of PKD2L1 function.

**Interactions between different domains**. TRP channels can be stimulated by diverse impetuses. For example, in TRPV1, the external signals are transmitted to the pore domain upon the binding of capsaicin to the S4–S5 linker and then mediate the gating process[22,29,32,39]. Alternatively, in NompC, a putative mechanosensitive TRPN channel, movements from cytoskeleton proteins cause its gate to open, specifically because of sequential allosteric transitions involving from the amino-terminal ankyrin repeats to the pore domain[40,41]. Generally, the pore domains of TRP channels will undergo conformational changes in response to structural variations from neighboring domains, which are in turn triggered by either mechanical stimuli or ligand signaling[1]. We, therefore, analyzed such inter-domain interactions in PKD2L1.

In PKD2L1, we noticed that the pore domain has close interactions with not only polycystin domains but the VSDs as well. Between the pore S5-PH-S6 and the polycystin domain, two groups of interactions exist at the interface. One group is composed of a pair of hydrogen bond formed by G260/R534 and a cation–π interaction between W259 and R534. The other group contains only one pair of hydrogen bond formed by T130/T501. The residues W259, G260, and T130 are from the polycystin domain while the residues R534 and T501 are located at knee points of S5-PH-S6 from an adjacent subunit. These structural features of interfaces are conserved in PKD2 and can be visualized in the closed-state PKD2L1 model (Fig. 5a, b).

Moreover, the pore domain also interacts with VSD domains through π–π interactions between S4 and adjacent PHs. In particular, a pair of F447/Y491 T-shape π–π interaction and several hydrophobic interactions can be observed (Fig. 5c, d). In addition, it seems that because the PKD2L1 S4–S5 linkers are flexible (they have invisible cryo-EM map density)[42], the S4

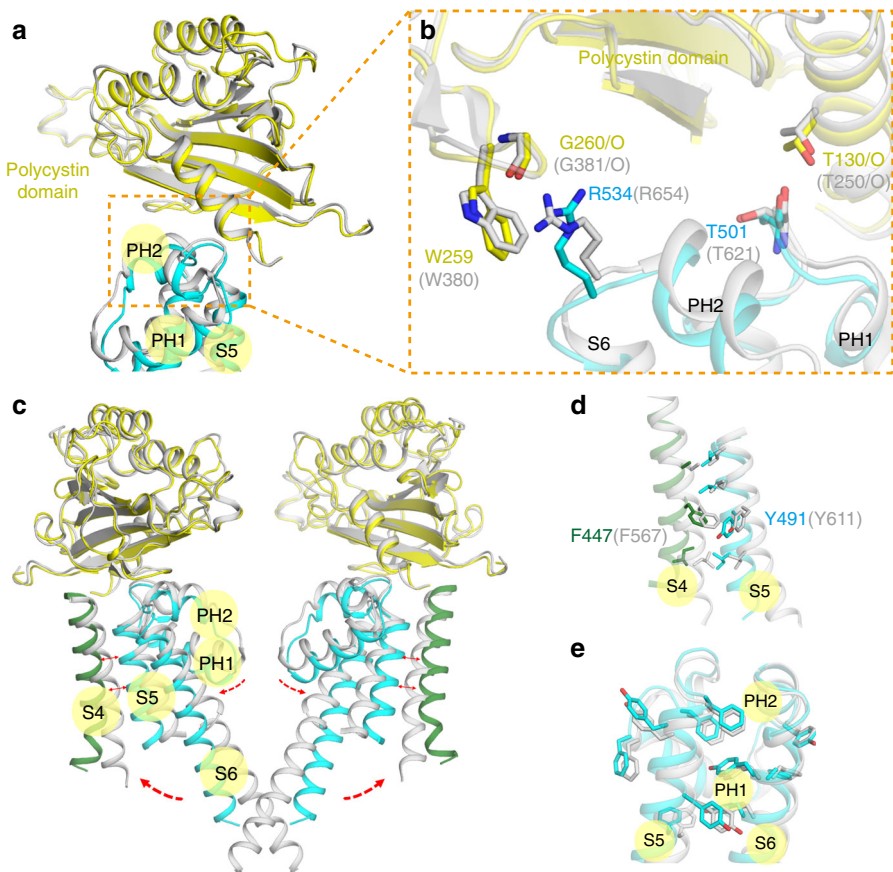

**Fig. 5** Interactions between pore domains and neighboring domains modeled in PKD2L1. **a** The upper gate is persistently close to the polycystin domain in both closed and open states. For visual clarity, only one diagonal protomer is shown. The polycystin domain (yellow represents PKD2L1 and gray for its homologous closed-state model) placed on top is from an adjacent protomer, depicting a close interaction with the pore domain. **b** The polycystin domain and the pore interact at two pivots, R534/W259 and T501/T130, around which the whole structure of the pore could swing forward and back. **c** The conformational changes pertaining to the pore regions (S5-PH-S6) between the PKD2L1 structure (cyan) and the homologous model (based on the PKD2 structure) representing its closed state (gray). For visual clarity, only two diagonal protomers are shown. The polycystin domains (yellow in the PKD2L1 structure and gray in the homologous model) placed on top and the S4 helix are from adjacent protomers, ensuring a close interaction with these pore domains, as shown in **d**. The red dotted arrows indicate regional shifts of the pore domain from a closed to open state. Reciprocal arrows between S4 and S5 are explained as in panel **d**. **d** Interactions between the S4 helix and an adjoining S5 helix guarantee that S5 sways together with S4 in the same direction. **e** Numerous aromatic residues may restrict the dynamics of each pore helix and maintain the related structure as a rigid entity

coupling motions in PKD2L1 may not transduce to S5 through the S4–S5 linker (Fig. 1b, c). The decoupled linker was also observed in a recently published KCNQ1 structure[43].

Collectively, the pore domain could interact with both the polycystin domains and the VSDs. Such interactions provide a connection between the pore and its neighboring domains, thus offering insights into a mechanism for cooperativity between adjacent domains and protomers. It is notable that all the residues involved in the interactions discussed above are conserved in PKD2 as well, thus being modeled and envisioned in the corresponding closed-state PKD2L1 model structure. This phenomenon indicates that the two proteins might share a similar signal transduction paradigm to some extent. These extensive pore domain interactions related to the pore domain might play crucial roles in the regulation of channels.

## Discussion

Electrophysiological studies have shown that PKD2L1 can respond to voltage stimuli as well as some exogenous or physiological modalities[7,26]. Meanwhile, the 6-TM TRPP subfamily proteins PKD2L1 and PKD2 harbor positively charged residues and 3₁₀ helical conformations in S4 (Fig. 3a and Supplementary

Fig. 9), both of which are hallmark features of VGICs[44]. These observations imply that the VSD's conformation may change as the membrane potential oscillates.

With the C-terminus truncated off from our PKD2L1 proteins, we intend to capture an open-conformation structure for the elucidation of potential mechanisms with consideration of several aspects. First, this construct (residues 64–629) shows a higher channel activity compared to wild-type PKD2L1 (Fig. 1e–h). Especially, we found that the decay phase of the EF-hand truncation was significantly increased than WT PKD2L1. The deletion of negative regulatory C-terminus (EF-hands domain) may attenuate the inactivation process to make the channel easier to stay in the open state. It is also noted that the wild-type and the truncation variant have similar localization patterns, which indicate no obvious difference in protein trafficking and have been validated by confocal fluorescence imaging (Supplementary Fig. 4c). These together might explain why we have a higher possibility in catching an open-state conformation of the proteins without exerting external stimuli to activate them. In addition, we used EDTA in our lysis buffer to chelate all divalent cations during sample preparation, and according to previous functional[7] studies, it is demonstrated that $Ca^{2+}$ from the intracellular side could induce channel inactivation. Hence the $Ca^{2+}$-free

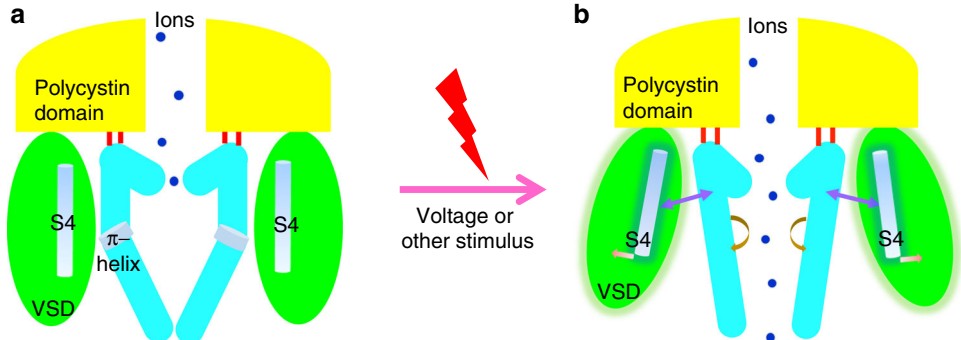

**Fig. 6** Hypothetical model for PKD2L1 gating mechanisms. Mechanisms of gating for PKD2L1 channels can be described as shown in the schematic. The closed-state of PKD2L1 depicted by the homologous model of PKD2 structure (**a**) implies the non-conductive state blocked hypothetically by sodium ions (marine dots). PKD2L1 represents the conductive state (**b**). The polycystin domain remains relatively static (a slightly outward eversion) during the gating process independent of the state. When the channel switches from closed to open in response to voltage or other physiological stimuli, its pore coupled with VSD motions dilates both the lower gate and the upper gate by an iris-like (for S6) and clock-like (for VSD) rotation, respectively

environment during structure resolving provided favorable conditions for the maintenance of open state. Overall, the truncated channel and $Ca^{2+}$-free conditions may help us gain the chance to obtain higher fraction of open-state particles to successfully get an open-state structure.

Structural determination of PKD2L1 and PKD2 in distinct states provides a unique opportunity to reveal the potential gating mechanism of PKD2L1. Through the conformational changes between the closed-state homologous PKD2L1 model and our open-state cryo-EM PKD2L1 structure, we propose the working mechanism of PKD2L1 (Fig. 6 and Supplementary Movies 1, 2, and 3). On perceiving an external stimulus, the R407 residue in the S3–S4 linker (Fig. 4c, gray) could be triggered and repositioned to a new conformation (Fig. 4c, green). This transfer disrupts the inner stability of the S3–S4 linker maintained by cation–π interactions between the arginine and a tryptophan (i.e., R407/W434). This leads to a conformation in which the S3 segment is stretched centrifugally and the S4 segment is translated extracellularly by only half of a helical turn, in comparison with respect to the whole structures (Fig. 4a). It appears that the VSD turns slightly around the axis of cation–π interactions between the S3–S4 linker and the polycystin domain (i.e., R199/Y425, Fig. 4b). The disturbance of the S3–S4 linker's cation–π interaction (Fig. 4c) as well as the absence of another oligosaccharide chain (Supplementary Fig. 6b) is accountable for the conformational changes of the S3–S4 linker and VSDs (Fig. 3b–d).

Our structural analysis also indicates that VSDs could couple to the pore gates. When the two states of PKD2L1 are superimposed, we closely analyzed the potential coupled motions arisen from VSD that are associated with pore domains' movements. It appears that the S4 helix drives the adjacent S5 helix from another protomer moving outwardly, owing to the VSD/pore domain π stacking interactions previously described (Fig. 5d). Moreover, there are multiple aromatic residues that could form extensive π–π interactions within the upper gate region (Fig. 5e). These multiple π-stacking interactions are very strong, and in some occasions can survive even in 2% sodium dodecyl sulfate[45,46]. Thus, these interactions may enable each pore domain protomer (S5-PH-S6) to travel as a rigid structure and may keep the upper gate relatively stable with respect to S6 during the gating process (Fig. 5c). Consequently, the entire upper gate region rotates like a clock pendulum swinging about 7° outwardly around the axis fixed by two pivots (R534, T501) at the interface of the polycystin domain/upper gate described above (Fig. 5a, b). The homodromously counterclockwise rotation of the

PH and the S6 helix dilates the upper gate, causing it to open (Figs. 5c and 6, and Supplementary Movie 3). Among TRP channels, it seems to be a common phenomenon that the upper gate undergoes remarkable conformational changes during the activation process[22,47,48]. Our structural analysis also supports this idea, depicting an unexpected conformational change of the PHs.

S6 segments undergo axial rotation unveiled by conformational transitions from the closed-state PKD2L1 model and the open-state cryo-EM structure, whereas the lower-gate dilates from 4.9 to 8.3 Å (Supplementary Movie 2). A state-dependent motion of the lower gate is suggested by the structural comparison between the two states, whereby L557 rotates away from the central pore, while I560 rotates in. All the voltage-gated potassium channels contain highly conserved glycine or proline residues in S6, which facilitate the hinge bending in S6, which is essential to opening the S6 bundle-crossing gate[49]. However, as TRPP subfamily channels all lack an equivalent glycine or proline in S6, we hypothesize that a π-helical configuration in the middle of S6 serves as a regulatory motif that is essential for gating. It is likely that S6 transits from a high-energy π-helix to a low-energy α-helix through the mediation of VSD movements, thus arousing the outward twisting of S6 and bringing about a dilation of the lower gate. A similar phenomenon can also be observed in $Na^+$ channels, as the open $EeNa_v1.4$ adopts α-helical S6 fragments compared with the closed $Na_vPaS$, which has a π-helix structural feature[50].

Interestingly, we observed similar gating mechanisms in the closed and open structure of TRPMLs, which were published recently[34–37]. In TRPML1, the agonists bind to a hydrophobic cavity within S5-P-S6 and the conformational changes triggered are associated with distinct dilations of its lower gate, together with a slight structural movement of PH1 (Supplementary Fig. 7a)[36]. To understand potential gating mechanisms, we, therefore, analyzed the hydrophobic cavities on both TRPML1 and PKD2L1 and found both cavities to be very similar structurally (Supplementary Fig. 7b–d). This leads to the speculation that some unknown ligands could bind to PKD2L1 and modulate it. In ligand-regulation of TRPML channel, the synergistic movement of the VSD and channel pore can be mutually observed, which interestingly, is similar to our proposed model (Fig. 6). All these features suggest us a common regulatory mechanism that may be shared in group II TRP channels.

Meanwhile, the recently published TRPML3 structures at pH 7.4 and pH 4.8 condition also deepen our understanding on $H^+$ regulation mechanisms[37]. Comparing the two conformations of

TRPML3, the luminal loops exhibit a large conformational change and seal the pre-pore valley when pH turns to 4.8. The conformational change then spreads to S1 and subsequently to S2, S3, the S4–S5 linker, S5, and S6, leading to an inhibited state when the luminal pH turns from neutral to acidic. (Supplementary Fig. 8a–c). The luminal loop of PKD2L1 shares significant sequence similarity and polar residues are conserved with that of TRPML3 (Supplementary Fig. 8e). With respect to PKD2L1's off-response mechanism and its assigned $H^+$ sensitivity, it could conceivably be hypothesized that $H^+$ might also transduce a signal from the luminal loop to the VSD and to the channel pore in PKD2L1 (Supplementary Fig. 8d). Thus the potential off-response mechanism for PKD2L1 may be that protons can bind to or protonate the negative charged residues to trigger the inhibition. Once the acid is withdrawn, inhibition can be transiently released to induce a large inward current (off-response).

Nevertheless, we cannot exclude the possibility that external signals can be transited through other domains, such as polycystin domains, onto the pore region, as they also have interfaces as discussed above. This is supported by Wilkes et al.[23], who suggested that PKD2 activation involves the conformational changes of both polycystin domains and C-terminal domains. However, the structural differences between the two states they reported are relatively small and their resolutions are barely enough to specifically identify lipids and side chains. Therefore, this hypothesis should be considered with cautions. The cryo-EM structure of the ion channel protein involved in renal and retinal development described herein may provide a basis for understanding the function of PKD2L1 in normal development and disease. However, a faithful understanding of PKD2L1 function and gating mechanisms, therefore, awaits additional atomic models in various conformations with functional validations.

## Methods

**Molecular biology**. *Mus musculus* PKD2L1 [GenBank: A2A259] was provided by Dr. H. Matsunami (Duke University). For structural studies, the truncated construct of mouse PKD2L1 (64–629) with an N-terminal triple Flag-tag (DYKDHDGDYKDHDIDYKDDDDK) and a C-terminal Strep-tag® II (WSHPQFEKGGGSGGGSGGGSAWSHPQFEK; from IBA GmbH) was also subcloned into a pCAG vector[51]. All segments subjected to PCR were verified by sequencing (All primer sequences used in this study are in a Supplementary Table 3).

**Transient expression of truncated mutant proteins**. The HEK 293F cell line (Invitrogen) was cultured in SMM 293T-I medium (Sino Biological Inc.) at 37 °C under 5% $CO_2$ in a Multitron-Pro shaker (Infors; 130 rpm). When cell density reached $2–2.5 \times 10^6$ cells per ml, PKD2L1 (residues 64–629) plasmids were transfected into the cells. For one liter of cell culture, 1.5 mg plasmid was premixed with 4 mg 25-kDa linear polyethylenimines (PEIs) (Polysciences) in 50 ml fresh medium for 15–30 min before transfection. Transfection was initiated by adding the mixture into cell culture and incubating for 15 min. Transfected cells were cultured for 48–60 h before harvesting.

**Whole-cell electrophysiology**. We performed whole-cell recordings in HEK293T cells at room temperature. Electrodes were pulled and heat-polished, resulting in 1–3 MΩ resistances before using. Whole-cell signals were acquired and analyzed by Axophtch 200B amplifier and the pCLAMP system (Molucular Devices). The Rapid Solution Changer (RSC-200) was applied for fast exposure of acid or high $Ca^{2+}$ solution. Bath solutions were perfused into the recording chamber with Valve Commander ALA-VM4 (ALA Scientific Instruments). The pipette solutions contained 140 mM KCl, 10 mM HEPES, 0.5 mM EGTA, at 290 mOsm adjusted with glucose and at pH 7.4 adjusted with KOH. The extracellular solutions for Off-response contained 135 mM NaCl, 5 mM KCl, 10 mM HEPES, and 2 mM $CaCl_2$, at ~300 mOsm adjusted with glucose and at pH 2.5 or 7.5 adjusted with HCl and NaOH. The extracellular solutions for $Ca^{2+}$-response contained 100 mM $CaCl_2$, 10 mM HEPES, at ~300 mOsm adjusted with glucose and at pH 7.5 adjusted with TEAOH. Data were analyzed in Clampfit (Molucular Devices, USA), Origin8 (OriginLab, USA) and Excel (Microsoft, USA). Standard error of the mean (S.E.M.) and student *t*-test (two-tailed, criteria of significance: $p < 0.05$, denoted as *; $p < 0.01$ as **; or $p < 0.001$ as ***) were calculated when applicable.

**Confocal microscopy fluorescence imaging**. Experiments were carried out in HEK293 cells expressing wild-type PKD2L1 or PKD2L1 (residues 64–629), each fused with a C-terminal YFP-tag and both expressed in complex with PKD1L3. The confocal fluorescence imaging experiments were performed with a ZEISS laser scanning confocal microscopy (LSM710). Data were collected and analyzed by ZEN 2012 Light Edition software.

**Purification of truncated mutant proteins**. For cell harvesting, transfected cells were centrifuged at $800 \times g$ and resuspended in lysis buffer containing 20 mM HEPES, pH 7.5, 150 mM NaCl, 10% glycerol, 5 mM EDTA and protease inhibitor cocktail (Amresco; 2 µg/ml aprotinin, 2 µg/ml leupeptin, 2 µg/ml pepstanin). The suspension was frozen by liquid nitrogen and stored at −80 °C until further operations. When purifying the protein, the thawed suspension was homogenized with 1 mM phenylmethylsulfonyl fluoride (PMSF) added. Then 2% dodecyl maltoside (DDM), 0.5% soybean lipids (Sigma), 0.4% cholesterol hemisuccinate (CHS) (Anatrace) was supplemented and the mixture was incubated for 1.5–2 h at 4 °C. After ultra-centrifuged at $18,700 \times g$ for 40–60 min, the supernatant was applied into anti-Flag M2 affinity gel (Sigma) for three times at 4 °C by gravity. The resin was rinsed eight times, 5 ml buffer containing 20 mM HEPES, pH 7.5, 150 mM NaCl, 10% glycerol and 0.06% digitonin (Sigma) per time. Target protein was then eluted with wash buffer plus 300–400 µg/ml Flag peptide (Sigma). The elution from Anti-Flag M2 column was loaded to Strep-Tactin resin (IBA company) for 1 h at 4 °C and then, the resin was washed extensively by the same wash buffer as above. Finally, the target protein PKD2L1 (residues 64–629) was eluted with wash buffer plus 5 mM D-Desthiobiotin (IBA). This final protein eluent was concentrated by a 100-kDa cut-off Centricon (Millipore) and further purified by Superose-6 column (GE Healthcare). The peak fractions were mixed with amphipols at 1:5 ratio (w/w), incubated overnight at 4 °C. Detergent was removed with Bio-Beads SM-2 (Bio-Rad; 4 °C for 2 h, 50–100 mg per 1 ml protein/detergent/amphipol mixture) added. Bio-Beads were eliminated by disposable needles and excess amphipols were removed by Superose-6 column (GE Healthcare) in a buffer composed of 20 mM HEPES, pH 7.5, 150 mM NaCl. The peak corresponding to tetrameric PKD2L1 channels was collected for cryo-EM analysis afterwards. This expression and purification strategy gives a typical yield of 0.02 mg homogeneous PKD2L1 (residues 64–629) with a N-terminal triple Flag and a C-terminal Strep II tag for every liter of HEK 293F cell culture.

**Electron microscopy grid preparation**. Aliquots (4 µl) of purified PKD2L1 (residues 64–629) at a concentration of approximately 5 mg/ml were placed on glow-discharged holey carbon grids (Quantifoil Cu 300 mesh, R1.2/1.3) which were glow discharged for 30 s (mid) after 2 min evacuation. The grids were blotted for 3.5 s and flash frozen in liquid ethane cooled by liquid nitrogen using Vitrobot Mark IV (FEI). All set grids were retained in liquid nitrogen until data collection.

**Cryo-EM data acquisition**. Grids of PKD2L1 (residues 64–629) were transferred to a 300 kV FEI Titan Krios TEM equipped with a FEI Falcon II direct electron detector and a Cs corrector. The Cs was adjusted lower than 10 µm. A total of 10,217 movie stacks were collected at a nominal magnification of 75,000 × (effective pixel size being 0.88 Å at the object scale) with the defocus ranging from −1.5 to −2.9 µm. Data collection was accomplished under low-dose conditions using automated software AutoEMation II (developed by Jianlin Lei)[52]. For each micrograph stack, 38 frames were collected, with a total electron dose at approximately 60 e⁻/Å2 with an exposure time of 2.25 s. The stacks were first motion corrected with MotionCorr[53] and the output stacks were further motion corrected with MotionCor2[54]. Dose weighting was performed concurrently[55]. Gctf[56] was used to estimate the defocus values.

**Cryo-EM image processing**. A representative diagram of image processing procedures on PKD2L1 is presented in Supplementary Fig. 2. All in all, 8251 good micrographs were manually selected based on their Thon rings and contaminated conditions, and 2,659,411 particles were automatically picked through RELION 1.4[53]. C4 symmetry was applied in all three-dimensional classification and refinement steps unless specifically noted. The map of PKD2 obtained from Shen et al. was low-pass filtered to 60 Å and was used as the initial model. All of the particles were first subjected to global angular search three-dimensional (3D) classification using RELION 2.0[54] with one class and step size of 7.5°, and then to 3D classification with five classes and 3.75˚ local angular search step. The local angular search 3D classification was executed multiple times, during which the input was from different iterations of the global angular search 3D classification. At this stage, 870,465 good particles were combined and further subjected to 3D auto-refinement. The output from 3D auto-refinement was then filtered to 60 Å and used as the reference for 3D classification with skipping alignment in the next step. In all, 109,470 good particles were obtained and further exposed to 3D auto-refinement, after which the resolution was 3.34 Å, following the gold-standard FSC 0.143 criterion[27] with a high-resolution noise substitution method. To obtain better density information in the S6 region, particles were further subjected to local search 3D classification with five classes and with skipping alignment. Then after applying a local mask to selected particles and redo auto-refinement and post-processing,

Finally, 22,296 good particles were obtained and subjected to the final resolution was 3.38 Å with better density information in the S6 area.

**Model building and structure refinement**. For model building of PKD2L1, the structure of PKD2 (PDB code: 5T4D) was used as the starting model and was fitted into PKD2L1 EM maps with UCSF Chimera[57]. Model building was performed in COOT[58]. De novo model building was performed subsequently. Sequence assignment was guided mainly by bulky residues and the chemical properties of amino acids were considered during model building. A poly-Ala model was built in some areas where the resolution was insufficient for side-chain assignments.

For structure refinement, we used the phenix.real_space_refine application of PHENIX[59] in real space with secondary structure as well as geometry restraints in order to prevent structure overfitting. To monitor overfitting of the overall model, we refined the model in one of the two independent maps from the gold-standard refinement approach and assessed the refined model against the other map[60] (Supplementary Figs. 1 and 3). The final model was evaluated using MolProbity[61]; statistics of the 3D reconstruction and model refinement can be found in Supplementary Table 1.

**Data availability**. The cryo-EM maps of the PKD2L1 structure have been deposited in the Electron Microscopy Data Bank (EMDB) with the accession code EMD-6877. The atomic coordinates for the corresponding model has been deposited in the Protein Data Bank (PDB) under the accession code 5Z1W. Electrophysiology data including all other data supporting the findings of this study are available from the corresponding authors upon reasonable request.

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

## Acknowledgements

We thank Dr. Xiaomin Li, Xiaofeng Hu, and Xiaomei Li for their technical supports. We thank Dr. Yigong Shi, Dr. Nieng Yan, Dr. Jiawei Wang for critical discussions. We thank the Tsinghua University Branch of China National Center for Protein Sciences (Beijing) for providing the facility support. We also want to thank Dr. Virginia Burger, Dr. Guanghui Yang, and Xiaolian Shen for their help in polishing words and phrases. The computation was completed on the "Explorer 100" cluster system of Tsinghua National Laboratory for Information Science and Technology. This work was supported by funds from China's Ministry of Science and Technology (grants 2013CB910602), National Natural Science Foundation (grants 31400631, 81370784, 31370822, and 21778034) and National Key R&D Program of China (2016YFA0501102).

## Author contributions

T.W. conceived the project. T.W. and Q.S. designed experiments. Q.S., F.H., Y.L., and X. G. prepared all construct variants. Q.S. and F.H. purified protein and prepared the EM samples. Q.S., F.H., Y.Z., and J.L. conducted the cryo-EM data collection. Y.Z. performed image processing, analyzed EM data and built the model, and C.Y. and Q.Z. also contributed. C.M., S.Y., A.S., Y.Z. and X.L. participated in the initiation of this project. Y.Z. and Y.L. participated in the manuscript preparations. X.L. and Y.L. designed and performed all electrophysiology. T.W., Q.S., F.H., Y.L., and X.L. wrote the manuscript.

## Additional information

**Competing interests:** The authors declare no competing interests.

