## [Peer Review File(PDF 254 kb) · Nature Communications]

Reviewers' comments:

Reviewer #1 (Remarks to the Author):

TRPP cation channels play crucial roles in cell physiology. Recently, the structure of the PKD2 protein has been solved, which greatly advanced our understanding of this functionally important protein. However, due to the fact that all the structures reported are all in closed conformation, the molecular mechanism of channel gating is still unsolved. In the current manuscript, Su et al. report a high-resolution cryo-EM structure of the homologous PKD2L1 channel. Although its overall layout is very similar to the reported PKD2 structure, the wider opening at the selectivity filter and the lower gate regions indicate a possible opening conformation of this structure. The comparison between this structure and the closed TRPP2 structure led to interesting analysis that may reflect the molecular mechanism of channel gating. This work will be of interest to others in the community and the wider field.

1) To avoid potential misleading by this structure, the authors should provide evidence to show that the truncated channel is still functional by doing electrophysiological measurements on channel activity.

2) It will be ideal to perform functional studies to confirm what is found in the structure to make this study complete. For example, experiments can be done to show mutating some of the key residues found to be crucial for gating affects channel activity.

3) Page 4, line 91, the authors claimed "...We found that only PKD2L1 forms a functional homotetrameric channel when heterologously expressed on the plasma membrane." This is not true since TRPP2 channel currents have been reported in many publications, from as early as Gonzalez-Perrett et al.'s PNAS paper in 2001 to the recent Arif Pavel et al.'s PNAS paper in 2016 with the GOF TRPP2 mutant.

4) The authors claimed that the pore domain is composed of four swapped S5-PH-S6 fragments. Although most likely this is the case, however, without seeing the S4-S5 linker in the structure, the reviewer suggests cautious interpretation of this domain arrangement.

5) The closed and open structure of TRPML1 is published recently. Due to the similarity between TRPP and TRPML channels, it will make sense to compare their potential gating mechanisms based on the structures.

6) The asparagine 533 has recently been reported to be essential for its voltage-dependent inactivation (Shimizu et al., 2017). It will be interesting to analyze any potential mechanism based on the structure.

7) Low pH has been shown to activate the PKD2L1/PKD1L3 complex through an off-response mechanism. The H⁺ sensitivity has also been assigned to the PKD2L1 homomeric channel (Hussein et al., 2015). By analyzing the current structure, can any clue be found on its pH sensitivity?

8) Page 10, line 211, "there is no obvious conformational change of the mucolipin domain in response to stimuli of different pH." This is also true in the ligated-gated TRPML1 channel (Schmiege et al., Nature, 2017).

9) It is easier to believe that ion channels should stay mostly in the closed state without receiving activation stimuli. The authors should discuss the possible reason of the channel was locked in the

open state during sample preparation. Do they also saw closed channels in cryo-EM images but their structure are not solvable?

10) Supplementary Fig. 1e. No black and green curves described in the legend are shown in the figure. The blue curve shown in the figure is not defined in the legend.

11) Supplementary Fig. 1f. The structures shown are too crowded to be distinguished from each other.

12) Supplementary Fig. 4f does not belong to the content of this figure.

Reviewer #2 (Remarks to the Author):

In the manuscript "Cryo-EM structure of polycystic kidney disease-like channel PKD2L1" Su et al. describe the structure of the TRP channel PKD2L1 by means of cryo EM and single particle analysis. The authors achieved a good resolution of an open conformation of this membrane protein. Comparisons to published TRP channels in closed conformation allowed to the authors to hypothesize about the mechanism of gating.

Even though several high-resolution cryoEM structures of TRP channels have already been solved the contributed high resolution structure is a worthy addition. The authors appear to have done a solid job on the cryoEM and single particle analysis. The Figure quality is sufficient but the manuscript writing style could be improved to improve clarity during reading I also feel that it is slightly to long. A few comments have to be further addressed before publication can be considered.

TRP channels usually exhibit a fourfold symmetry that can be used to amplify the signal and obtain improved maps. In the extended data table the authors also highlight that this symmetry was used but I was unable to find it in the main text/methods section. Please clarify.

The data was collected using the Falcon 2 detector provided by FEI (now Thermo). To the best of my knowledge only the late generation models of these detectors were able to perform movie acquisition but only 7 frames – not 38 as stated. Greg McMullan (LMB Cambridge) however, developed a hacked version which would then enable movie acquisition using a more or less home built device. The authors need to clarify which version they have used, or how they were able to obtain such movies and if not developed by them cite the person.

The SupplementFigure 1 needs to be updated. 2D classaverages are too small to visualize anything and this is even more severe in the raw micrograph B. I suggest to show a much larger micrograph with a further magnified inset so that the quality of the preparation can be assessed.

The authors further write that the good micrographs were selected – based on what criteria?

For single particle reconstruction 2.5 mio. particles were selected from which only 10% were considered good and only 75k ended up in the final model. While it is commonly accepted that most, or at least a large fraction, of selected particles fail to provide a reasonable density map these number seem very high.

Especially, in light of the newly found conformation the authors need to explain what happened with the remaining particles (especially the 175k that were considered good) in more detail. Where different – closed?- conformations present in this dataset - or would an addition of these particles

reduce the resolution? Is it possible that the majority of particles were indeed topviews and therefore did not contribute to improve the resolution? Please explain.

Response to Reviewers Comments

Reviewer #1:

This reviewer commented favorably on our manuscript and raised twelve specific points for us to address.

1. To avoid potential misleading by this structure, the authors should provide evidence to show that the truncated channel is still functional by doing electrophysiological measurements on channel activity.

We agree with the reviewer that the truncated PKD2L1 needs to be functionally validated. As the pore-forming subunit, PKD2L1 proteins can function either as homotetrameric channels or the heteromeric channels in complex with PKD1L3. PKD2L1/PKD1L3 expressed on the membrane of mammalian cells can produce acid-evoked response or off response (Inada et al. *EMBO Rep.* 2008; Hussein et al. *Scientific Reports* 2015; Hu et al. *Cell Reports* 2015), which has been considered as the potential mechanism underlying proton sensing in neurons. PKD2L1 can also produce calcium response, *e.g.*, from homomeric PKD2L1 expressed in oocytes (Chen et al. *Nature* 1999) or from heteromeric PKD2L1/PKD1L3 in HEK293 cells (Hu et al. *Cell Reports* 2015). Both of these unique responses to extracellular stimuli represent physiologically relevant functionalities of PKD2L1. Based on the above evidence, we decided to examine the functions of truncated PKD2L1 co-expressed with PKD1L3 in HEK293 cells, in the context of the off response and calcium response.

According to the results assayed with whole-cell patch clamping (Fig. 1e-1h), truncated PKD2L1 (residues 64-629), whose structure we resolved, preserves all the major features established from the wild-type PKD2L1 in full length. In fact, truncated PKD2L1 exhibited even more pronounced currents, especially to calcium stimuli, potentially due to elimination of the inhibitory EF-hands on the C-terminus (Hu et al. *Cell reports* 2015). We conclude that the truncated PKD2L1 (64-629) is still functional as validated by its acid-evoked off response and calcium-influx induced calcium response.

The main text was revised accordingly (page 6, line 125).

2. It will be ideal to perform functional studies to confirm what is found in the structure to make this study complete. For example, experiments can be done to show mutating some of the key residues found to be crucial for gating affects channel activity.

We agree with the reviewer that it would be ideal and more complete if further functional studies were performed. As suggested, we conducted mutagenesis to validate one major finding of this study that both the outer pore and lower gate are crucial to PKD2L1 gating. According to the new insights from the structure, both

the outer pore and the inner gate are subject to dilation, which is coupled with dynamic changes in channel conformations. We then constructed G522L or I560F mutant channels, which carry mutations at the two major constriction sites (G⁵²² at the outer pore and I⁵⁶⁰ at the lower gate) respectively (Supplementary Fig. 4). Such mutations severely impaired PKD2L1 gating, as proved by the loss of functions in response to stimuli of both modalities, potentially by side-chain extensions of Leu or Phe that may occlude ion conduction.

The main text was revised accordingly (page 9, line 203).

3. Page 4, line 91, the authors claimed "... We found that only PKD2L1 forms a functional homotetrameric channel when heterologously expressed on the plasma membrane." This is not true since TRPP2 channel currents have been reported in many publications, from as early as Gonzalez-Perrett et al.'s PNAS paper in 2001 to the recent Arif Pavel et al.'s PNAS paper in 2016 with the GOF TRPP2 mutant.

Point appreciated. We apologize for this oversight. We have amended the sentence on page 4, line 97.

4. The authors claimed that the pore domain is composed of four swapped S5-PH-S6 fragments. Although most likely this is the case, however, without seeing the S4-S5 linker in the structure, the reviewer suggests cautious interpretation of this domain arrangement.

Point accepted and appreciated. We should have been more concise and cautious to claim structural features with uncertainty. We have accepted the suggestion from the reviewer and have made this point clear in the revised manuscript on page 7, line 143.

5. The closed and open structure of TRPML1 is published recently. Due to the similarity between TRPP and TRPML channels, it will make sense to compare their potential gating mechanisms based on the structures.

We appreciate this insightful suggestion. Indeed, both TRPP and TRPML belong to group II TRP channels, and they share great similarities in their overall structures, including the transmembrane voltage sensing domains (VSDs), and the extracellular polycystin/mucolipin domains in particular (Shen et al. *Cell* 2016; Schmiede et al. *Nature* 2017). Also, both TRPML1 and PKD2L1 are subject to voltage activation or regulation, thus potentially sharing common voltage-gating mechanisms, which were analyzed in our study in details for PKD2L1. During dynamic changes of the structure upon activation, PKD2L1 resembles TRPML1 in the prominent feature of dual-pore dilation, indicative of channel gating at both the upper gate (outer pore) and inner pore (lower gate). Meanwhile, for TRPML, the agonists or activators have been successfully identified such as ML-SA1, which provide great help for its structural study by locking the channel at the ligand-bound state. However, lack of

such a ligand for TRPP restricts us from more explicitly exploring the open structure. Hypothetically, some unknown ligands could bind to the sites around the outer pore and activate PKD2L1 in a way similar to ML-SA1/TRPML1, to further support the notion that the group II TRP channels would share common gating mechanisms. We have revised the text accordingly in the relevant parts of the main text (page 18, line 394 and Supplementary Fig. 7).

6. The asparagine 533 has recently been reported to be essential for its voltage-dependent inactivation (Shimizu et al., 2017). It will be interesting to analyze any potential mechanism based on the structure.

Point appreciated. Even though it is not easy to fully interpret the role of asparagine 533 in PKD2L1's voltage-dependent inactivation based on our structure, we still try to postulate a potential mechanism in response. However, we would like to refrain from detailed discussions in the main text.

As found by Takahiro Shimizu in 2017 *FEBS Open Bio.* paper, the N533Q mutant of PKD2L1 generates a greater outward current upon depolarization with no subsequent inactivation. Combined with the PKD2L1 structure that we have resolved, it is interesting to see that the side chain of N533 in the outer pore loop region is close to Q437 and N440 (located on S4), which indicates that they may interact with each other during the process.

A potential mechanism for PKD2L1's voltage-dependent inactivation might be as the following: First, the voltage changes from membrane influence the VSD domain, particularly S4 helices. Then, motions are further transmitted onto pore helices through the residues mentioned above. Our hypothesis remains to be validated by electrophysiological experiments in the future.

7. Low pH has been shown to activate the PKD2L1/PKD1L3 complex through an off-response mechanism. The H⁺ sensitivity has also been assigned to the PKD2L1 homomeric channel (Hussein et al., 2015). By analyzing the current structure, can any clue be found on its pH sensitivity?

Thanks for your comments. For PKD2L1, the proton sensitivity is manifested by the unique off response upon acid withdrawal. The mechanism underlying such acid-evoked off response is still unclear (Chen et al. *European Biophys.* 2016; Hu et al. *Cell Reports* 2015; DeCaen et al. *eLife* 2016). Because of its structural and functional similarities with TRPML, PKD2L1 may share some common mechanisms with TRPML regarding proton regulation.

For TRPML, the structures of different conformations have recently been published, providing more details on the gating mechanisms potentially shared by group II TRP channels (Chen et al. *Nature* 2017; Hirschi et al. *Nature* 2017; Schmiege et al. *Nature* 2017; Zhou et al. *NSMB* 2017). Structural and functional studies suggest that protons induce inhibitory effects on channels through protonation of acidic residues on extracytosolic domains, e.g., luminal loops. Meanwhile, the protonation could

also attenuate Ca²⁺ blockage of the channel, underlying proton activation or sensitization observed in experiments.

Notably, our PKD2L1 structure unveiled a similar luminal pore within polycystin domains (Supplementary Fig. 8e), inviting further investigations in comparison with TRPML (Chen et al. *Nature* 2017; Zhang *Protein Cell* 2017).

To this end, enlightened by our and other studies, both PKD2L1 and TRPML appear to follow similar principles in sensing the extracellular protons.

Please see the revised manuscript for details (page 18, line 405 and Supplementary Fig. 8).

8. Page 10, line 211, “there is no obvious conformational change of the mucolipin domain in response to stimuli of different pH.” This is also true in the ligated-gated TRPML1 channel (Schmiege et al., Nature, 2017).

Point accepted. This newly published work does help us understand TRPML channels more comprehensively, which can be regulated by not only pH but ligands as well. We have therefore amended this sentence in page 11, line 249.

9. It is easier to believe that ion channels should stay mostly in the closed state without receiving activation stimuli. The authors should discuss the possible reason of the channel was locked in the open state during sample preparation. Do they also saw closed channels in cryo-EM images but their structure are not solvable?

We appreciate the insightful comments. We agree with the reviewer on the statement that ion channels normally are more likely in the closed state without specific stimuli to activate the channel. However, in this particular case of PKD2L1, there are a few reasons to support that it is fairly possible to capture the open structure of the channels as we achieved in this study. First, as we mentioned in the answer to question 1, deletion of C-terminal EF-hands significantly enhanced channel activation by attenuating Ca²⁺-dependent inactivation or inhibition (Hu et al. *Cell Reports* 2015), manifested by the increase in current amplitude and the prolongation of the decay time (Fig. 1e-1h). Such gain-of-function effects should contribute to the higher fraction of channels in the open state overall in comparison to wild-type channels containing EF-hands. Second, PKD2L1 channels are featured with unique gating properties (including relatively high open probability and large single-channel conductance), and in particular potential activation by changes in calcium and pH. Both features might take actions to open a substantial number of the channels. Furthermore, in our practice, we used EDTA in the lysis buffer to chelate all divalent cations during sample preparation. The Ca²⁺-free environment favors channel activation by attenuating the Ca²⁺-dependent inactivation or inhibition (Hu et al. *Cell Reports* 2015; DeCaen et al. *eLife* 2016), and this is another potential factor to account for the open state we captured in our structure. Therefore, the truncated

channels and Ca²⁺-free conditions helped us obtain higher fraction of open-state particles to successfully resolve the structure indicative of channel activation. Besides, during image processing, we didn't see any closed channels in cryo-EM images. On the one hand, closed or open conformations were not discernible based on micrographs. On the other hand, we failed to witness a closed conformation in 2D or 3D classifications due to the limited proportion of closed-state particles. We have revised the main text accordingly (page 15, line 332).

10. Supplementary Fig. 1e. No black and green curves described in the legend are shown in the figure. The blue curve shown in the figure is not defined in the legend.

Point accepted. We erroneously described the legend for Supplementary Fig. 1e. A correction has been made and we have added a new Supplementary Fig. 1f showing FSC curves for cross-validation between the model and the cryo-EM maps in the revised manuscript. (Supplementary Fig. 1e and 1f)

11. Supplementary Fig. 1f. The structures shown are too crowded to be distinguished from each other.

Point accepted. With four PKD2 structures having been resolved, we intended to show that the overall architecture of PKD2L1 is identical to those of PKD2s, while it is still different to certain extent (especially the intracellular side) as well. We have replaced Supplementary Fig. 1f. with Supplementary Fig. 5c containing four panels. Supplementary Fig. 5c shows the structural alignments of our PKD2L1 to all four published PKD2 structures: 5T4D (gray), 5K47 (bluewhite), 5MKE (wheat) and 5MKF (pale cyan). All structures are superimposed holistically. PKD2L1 adopts a distinct conformation in comparison with all four PKD2 structures. (Supplementary Fig. 5c)

12. Supplementary Fig. 4f does not belong to the content of this figure.

Point appreciated. Supplementary Fig. 4f was the comparison of TRPV1's selectivity filter between two conformations, while the remaining panels were about polycystin domains. Therefore, we have rearranged Supplementary Fig. 4a-e into Supplementary Fig. 6a-e in the revised manuscript. We have prepared a new Supplementary Fig. 5 for structural and conformational changes of different structures, and kept the previous Supplementary Fig. 4f as the new Supplementary Fig. 5b.

We thank this reviewer for his/her time and constructive comments.

Reviewer #2:

This reviewer commented favorably on our manuscript and raised several points for us to address.

1. TRP channels usually exhibit a fourfold symmetry that can be used to amplify the signal and obtain improved maps. In the extended data table the authors also highlight that this symmetry was used but I was unable to find it in the main text/methods section. Please clarify.

Point accepted. It is clearer to explain the fourfold symmetry in the main text/methods section. We have provided the relevant information on page 5, line 104 and page 24, line 516 in the revised manuscript.

2. The data was collected using the Falcon 2 detector provided by FEI (now Thermo). To the best of my knowledge only the late generation models of these detectors were able to perform movie acquisition but only 7 frames – not 38 as stated. Greg McMullan (LMB Cambridge) however, developed a hacked version which would then enable movie acquisition using a more or less home built device. The authors need to clarify which version they have used, or how they were able to obtain such movies and if not developed by them cite the person.

We appreciate the reviewer's attention to technical details. As for the Falcon 2 detector, the reviewer's comment might be true more than one year ago. Fortunately, the Titan Krios TEM instruments we used to collect data in this paper had its FEI software upgraded to version 2.5 in July 2016 (by FEI), and the new version allows us to acquire up to 40 frames (~17 frames per second) without hacking. Our data was collected in December 2016 and January 2017.

Before July 2016, We used a hacked version developed by MRC in order to get more than 7 frames on TEMs equipped with Falcon 2 detector.

We have amended the related discussion on page 24, line 507 in the revised manuscript.

3. The Supplement Figure 1 needs to be updated. 2D classaverages are too small to visualize anything and this is even more severe in the raw micrograph B. I suggest to show a much larger micrograph with a further magnified inset so that the quality of the preparation can be assessed.

Point accepted. Both figures have been enlarged for better quality with a further magnified inset in the revised manuscript. (Supplementary Fig. 1)

4. The authors further write that the good micrographs were selected – based on what criteria?

Our selection criteria for good micrographs were directly based on the qualities of Thon rings and the conditions of micrographs.

Specifically, we obtained the Thon rings in our micrographs after the cryo-EM data were collected using a direct detection camera following implemented procedures of dose fractionation and motion correction (MotionCorr, developed by Xueming Li; Li, X. et. al. *Nature Methods* 2013). Then micrographs were manually screened through dosef_logviewer (also by X. Li) to guarantee the quality even if they were taken at different defocus levels. Finally, Thon rings of selected micrographs could extend to 3-4 Å on average in the Fourier power spectrum of almost every motion-corrected image.

At the same time, we also eliminated micrographs that were contaminated by ice or chemicals, or those that were distinguished by amorphous ice. Major text was revised accordingly (page 24, line 515).

5. For single particle reconstruction 2.5 mio. particles were selected from which only 10% were considered good and only 75k ended up in the final model. While it is commonly accepted that most, or at least a large fraction, of selected particles fail to provide a reasonable density map these number seem very high.

We would like to explain this big difference in terms of particle usage in details. Previously in the main text/method section, we made some mistakes about the particle numbers during data processing, though the flowchart in Supplementary Fig. 2 is correct. According to the flowchart, we have corrected the relevant details in the main text/method section (page 25, line 532).

After we selected good micrographs based on Thon Ring's quality, 2,659,411 particles were auto-picked using Relion-1.4. This data set included a great portion of bad particles, for example, from micrographs containing empty background area or ice contaminations. Due to its large data size, we chose to skip 2D classification, and performed another effective data-processing pipeline as shown in the flowchart (Supplementary Fig. 2). After particle auto-picking, the particles were first subjected to 3D classification with a step size of 7.5° global angular search, and then to 3D classification with 5 classes and 3.75° local angular search step. The local angular search 3D classification was executed multiple times, during which the input was from different iterations of the global angular search 3D classification.

Until then, 870,465 "good particles" were selected from each 3D classification and combined and further subjected to 3D auto-refinement. By adopting this method, we believed that a large fraction of bad particles (67.3% of all auto-picked particles) was excluded but still there were many "inappropriate" particles in the current data set. Therefore, we performed local search 3D classification with skipping alignments and 109,470 particles (4.1% of all auto-picked particles, 12.6% of "good particles") were selected, and the resolution reached 3.34 Å. During model building, we found that the S6 region of the 3.34 Å map was not good enough to build an atomic model, so we performed another local search 3D classification to discard bad particles and see whether the local resolution of S6 region could be increased. Finally, 22,296 particles

were selected from one category and the resolution of the structure was 3.38 Å after refinement, while more side chains could be seen in the S6 region. Therefore, we decided to build the final atomic model based on this 3.38 Å structure.

6. Especially, in light of the newly found conformation the authors need to explain what happened with the remaining particles (especially the 175k that were considered good) in more detail. Where different – closed?- conformations present in this dataset - or would an addition of these particles reduce the resolution? Is it possible that the majority of particles were indeed topviews and therefore did not contribute to improve the resolution? Please explain.

In our experiments, the ice of the cryo samples was relatively thicker than expected, and we did not have a serious particle orientation preference in our case (Supplementary Fig. 1c). The majority of particles were removed due to the flexibility of the S6 region and the quality of particles.

As stated, 109,470 particles (4.1% of all auto-picked particles, 12.6% of “good particles”) were remained after 3D classification, and the corresponding resolution was 3.34 Å. In this step, the removed particles were speculated to be relatively bad particles, the addition of which would reduce the resolution (to 3.67 Å).

From the 109,470-particle dataset, 22,296 particles were selected from a 3D classification with 5 classes. Although the resolution of 3.38 Å was a little bit lower than 3.34 Å, more side chains appeared in the S6 area of PKD2L1 in the 3.38 Å structure. As we explained to reviewer #1 in question 9, only one conformation was identified during the structure resolving. The other 4 classes were in similar conformation to our final structure, but with lower local resolution in the S6 region.

We thank this reviewer for his/her time and constructive comments.

REVIEWERS' COMMENTS:

Reviewer #1 (Remarks to the Author):

In this revised manuscript, the authors have answered all my concerns about the original manuscript. They have added functional data showing the truncated channel is functional. The manuscript is in much better shape now. However, after reading this new version, I still have some minor concerns, mainly about the manuscript writing.

1. Authors should make sure their citation is accurate and complete.

For example, in Line 83, authors cited three reports about the PKD1L3/PKD2L1 complex's function as sour taste and pH-dependent regulations. The authors should cite either the very first papers which brought out this concept, or all important papers, or just recent reviews. If they want to cite all major finding on this topic, the following reports should be on the list:

- 1) Ishimaru et al., Transient receptor potential family members PKD1L3 and PKD2L1 form a candidate sour taste receptor. PNAS, 2006
- 2) LopezJimenez et al., Two members of the TRPP family of ion channels, Pkd1l3 and Pkd2l1, are co-expressed in a subset of taste receptor cells. Journal of neurochemistry, 2006
- 3) Yu et al., Structural and molecular basis of the assembly of the TRPP2/PKD1 complex, PNAS, 2009
- 4) Kawaguchi et al., Activation of polycystic kidney disease-2-like 1 (PKD2L1)-PKD1L3 complex by acid in mouse taste cells. JBC, 2010

At the same time, they may want to bring out the controversial finding in the function of PKD1L3 as sour taste receptor.

Another example is the following sentence (line 85). The authors cited four papers when they talked about the function of PKD1L1/PKD2L1. However, the first paper they cited (Ref #18) was about PKD1L3/PKD2L1, and the second one they cited (Ref #19) was about PKD2L1 alone, instead of its complex with PKD1L3.

Please double check the other places.

2. The flow of the text and the citation on figures are messy in many places, making it difficult to follow. The authors should carefully check their figure citations to make sure they are cited at the right positions.

For example, line 165, I don't understand why Fig. 2C is cited here. The following several sentences after this sentence have no citation. So as a reader I can only assume that what the authors are talking about are in Fig. 2c. However, Fig. 2C does not have this information. Four sentences later, the authors cited Fig. 2d. I then found out that what they just talked about is actually shown in Fig. 2d. The sentence in line 192 needs a figure citation. The sentence in parenthesis belongs to figure legend instead of main text.

Line 213. Should the citation be 2c or 2e? The figure legend of 2e, instead of 2c, talked about the n-helix and α -helix.

3. Lines 330-333, the authors claimed that the higher activity of the shorter form of PKD2L1 could be the reason why they only found open-state conformation. However, the higher activity of the short-form PKD2L1 may also occur because of a better trafficking to plasma membrane instead of a higher single-channel open probability.

4. Lines 337-340, the authors said Ca^{2+} could induce channel inactivation so their EDTA-containing solution will help to keep the channel in open state. However, Ca^{2+} activate PKD2L1 channel as shown in Fig. 1e.

5. Figure legend of 2b in lines 749-752 belong to another place since residue number of PKD2 is not in grey here, and they are not shown after the slash. I guess it belongs to Fig 2c. At a right place, it should also be indicated that PKD2 and PKD2L1 structures are also shown in cyan and grey respectively. The same problem happens in the legend of Fig 3b. in lines 776 to 777. There is no residue name and number shown in this figure.

6. Lines 801, if these residues are not shown in the fig 2a, they should not be mentioned here and this sentence should be moved to the legend of Fig. 2b.

7. Fig. 6b. Maybe the authors should add some dots/ions in the lower pore region to show the channel is conducting ions through the opened pore.

8. I understand that the authors had used 100 mM Ca^{2+} to induce the PKD2L1 channel in their previous publication. However, 100 mM is a really high concentration and is way higher than the physiological concentration of Ca^{2+} . In fact the channel activity was induced by 5 mM Ca^{2+} in the original paper of Ca^{2+} -induced PKD2L1 channel activity published by Chen et al. in 1999. Can the authors explain why they have to use 100 mM Ca^{2+} ? Does this high Ca^{2+} solution cause any response of the untransfected cells (no negative control are shown in Fig. 1)?

Reviewer #2 (Remarks to the Author):

The authors have addressed all of my concerns in a very detailed and careful manner. I would accept the revision without further changes.

Response to Reviewers Comments

We thank the reviewers for their helpful comments, which provided us with critical inputs that enabled us to improve our manuscript. We have performed additional experiments and revisions as requested. This letter contains a full point-by-point response to the reviewers' comments. Please note that all modifications for this round of review are highlighted in yellow in the manuscript.

Reviewer #1:

This reviewer commented favorably on our manuscript and raised eight specific points for us to address.

1. Authors should make sure their citation is accurate and complete.

For example, in Line 83, authors cited three reports about the PKD1L3/PKD2L1 complex's function as sour taste and pH-dependent regulations. The authors should cite either the very first papers which brought out this concept, or all important papers, or just recent reviews. If they want to cite all major finding on this topic, the following reports should be on the list:

- 1) Ishimaru et al., Transient receptor potential family members PKD1L3 and PKD2L1 form a candidate sour taste receptor. PNAS, 2006*
- 2) LopezJimenez et al., Two members of the TRPP family of ion channels, Pkd1l3 and Pkd2l1, are co-expressed in a subset of taste receptor cells. Journal of neurochemistry, 2006*
- 3) Yu et al., Structural and molecular basis of the assembly of the TRPP2/PKD1 complex, PNAS, 2009*
- 4) Kawaguchi et al., Activation of polycystic kidney disease-2-like 1 (PKD2L1)-PKD1L3 complex by acid in mouse taste cells. JBC, 2010*

At the same time, they may want to bring out the controversial finding in the function of PKD1L3 as sour taste receptor.

Another example is the following sentence (line 85). The authors cited four papers when they talked about the function of PKD1L1/PKD2L1. However, the first paper they cited (Ref #18) was about PKD1L3/PKD2L1, and the second one they cited (Ref #19) was about PKD2L1 alone, instead of its complex with PKD1L3.

Please double check the other places.

Point accepted and appreciated. We should have been more concise and cautious to quote citations. We have accepted the suggestion from the reviewer and have revised this point in the revised manuscript in line 83, line 85 and lines 614-620.

2. The flow of the text and the citation on figures are messy in many places, making it difficult to follow. The authors should carefully check their figure citations to make sure they are cited at the right positions.

For example, line 165, I don't understand why Fig. 2C is cited here. The following several sentences after this sentence have no citation. So as a reader I can only assume that what the authors are talking about are in Fig. 2c. However, Fig. 2C does not have this information. Four sentences later, the authors cited Fig. 2d. I then found out that what they just talked about is actually shown in Fig. 2d.

The sentence in line 192 needs a figure citation. The sentence in parenthesis belongs to figure legend instead of main text.

Line 213. Should the citation be 2c or 2e? The figure legend of 2e, instead of 2c, talked about the π -helix and α -helix.

Point accepted and appreciated. We apologize for these oversights. We have amended these points in line 168, 170, 172, 196 and 216 respectively.

3. Lines 330-333, the authors claimed that the higher activity of the shorter form of PKD2L1 could be the reason why they only found open-state conformation. However, the higher activity of the short-form PKD2L1 may also occur because of a better trafficking to plasma membrane instead of a higher single-channel open probability.

We agree with the reviewer that in addition to a higher P_o (open probability), a better trafficking to plasma membrane could also lead to the higher activity of channels. In this particular case of truncated PKD2L1 (residues 64-629), we and other labs have reported that the truncations of the C-terminus containing EF-hands upregulate channel activities (Hu *et al.*, Cell Reports 2015). Such enhancement of channel functions can be manifested into two aspects: the current amplitude and the time constant of the decay phase. As shown experimentally in this work, both the amplitude and the decay time were significantly increased for the EF-hand truncation (Fig. 1e-h), in comparison with the WT channels. Notably, the kinetics of the decay phase, *e.g.*, as shown in the acid-evoked off response (Hu *et al.*, Cell Reports 2015), could be fully Ca^{2+} -dependent, reversely correlated with the current amplitude or the amount of Ca^{2+} entry. Further confocal imaging experiments confirm that the expression patterns of WT and truncation variant exhibit no difference (Supplementary Fig. 4c).

Taking together, we would like to conclude that the enhanced gating should make a significant contribution to the upregulation of channel activities that helped achieve

the open-state structure. And the contribution from potential membrane trafficking should not be dominant, if there is any, to the high channel activity in this study. We have amended the relevant information in page 15, lines 338-344 and Supplementary Fig. 4c in the revised manuscript.

4. Lines 337-340, the authors said Ca²⁺ could induce channel inactivation so their EDTA-containing solution will help to keep the channel in open state. However, Ca²⁺ activate PKD2L1 channel as shown in Fig. 1e.

We appreciate this insightful comment. It is true that Ca²⁺ plays dual roles in both activation and inactivation of PKD2L1 channels, but further details are needed to analyze each specific case to reconcile any potential confusion.

We would like to point out two aspects on this issue. First, thus far, Ca²⁺ is found to act only from extracellular space on PKD2L1 channels in ICE or Ca²⁺ response; in contrast, Ca²⁺ is generally considered to inhibit the channel from the intracellular side (Chen *et al.*, Nature 1999; Hu *et al.*, Cell Reports 2015; DeCaen *et al.* eLife 2016). Second, the (extracellular) Ca²⁺ activation has a transient nature, and the channels would go less active due to (intracellular) Ca²⁺ inactivation, the balance of which could account for the steady-state current of relatively high amplitude (a few hundred pA or even higher at -60 mV).

And our EDTA treatment of cell lysis thus would greatly help relieve the Ca²⁺ inactivation, unveiling the active state of the channels initially triggered by Ca²⁺ influx and/or other factors, similar to the case of acid-evoked off response where channels would stay open if intracellular Ca²⁺ is substantially reduced (Hu *et al.* Cell Reports 2015).

We have revised the text to include the above discussion (line 348-349).

5. Figure legend of 2b in lines 749-752 belong to another place since residue number of PKD2 is not in grey here, and they are not shown after the slash. I guess it belongs to Fig 2c. At a right place, it should also be indicated that PKD2 and PKD2L1 structures are also shown in cyan and grey respectively. The same problem happens in the legend of Fig 3b. in lines 776 to 777. There is no residue name and number shown in this figure.

Point accepted. We erroneously described the legend for Fig. 2b and 3b. Corrections have been made in the revised manuscript (lines 781-784 and lines 809-812).

6. Lines 801, if these residues are not shown in the fig 5a, they should not be mentioned here and this sentence should be moved to the legend of Fig. 5b.

Point accepted. We have adjusted the legend for Fig. 5a and 5b. Corrections have been highlighted in the revised manuscript in line 834.

7. Fig. 6b. Maybe the authors should add some dots/ions in the lower pore region to show the channel is conducting ions through the opened pore.

Point appreciated. We have updated Fig. 6b with some ions (marine dots) in the lower pore region to visualize the ion-conducting state of the open conformation.

8. I understand that the authors had used 100 mM Ca²⁺ to induce the PKD2L1 channel in their previous publication. However, 100 mM is a really high concentration and is way higher than the physiological concentration of Ca²⁺. In fact the channel activity was induced by 5 mM Ca²⁺ in the original paper of Ca²⁺-induced PKD2L1 channel activity published by Chen *et al.* in 1999. Can the authors explain why they have to use 100 mM Ca²⁺? Does this high Ca²⁺ solution cause any response of the untransfected cells (no negative control are shown in Fig. 1)?

We thank the reviewer for bringing up this critical point. We agree that Ca²⁺ of 100 mM is way beyond the physiological range; instead, using this high Ca²⁺ is mainly for the clarity of biophysical characterizations of this unique feature of PKD2L1. In the work first time reporting the Ca²⁺ response from mammalian expression of PKD2L1 (Hu *et al.* Cell Reports 2015), although our data were mainly based on 100 mM Ca²⁺ activation, we confirmed that the matter of concentration is just a reflection of the balance between the strength of the activation and that of the inactivation. With certain variants and/or under specific conditions, the response can be induced by Ca²⁺ of as low as 2 mM (Hu *et al.* Cell Reports 2015). So it is not imperative to use 100 mM Ca²⁺, which would not create any appreciable response from control cells (all data including negative control can be found in Hu *et al.* Cell Reports 2015), thus would not cause any other response of the untransfected cells. We have therefore replenished legends for Fig. 1. in line 764.

We thank this reviewer for his/her time and constructive comments.

Reviewer #2:

This reviewer commented favorably on our manuscript and has no further questions for us to address.

We thank this reviewer for his/her time and devotions.